# STABILITY BASED GENERALIZATION BOUNDS FOR EXPONENTIAL FAMILY LANGEVIN DYNAMICS

## ABSTRACT

We study the generalization of noisy stochastic mini-batch based iterative algorithms based on the notion of stability. Recent years have seen key advances in data-dependent generalization bounds for noisy iterative learning algorithms such as stochastic gradient Langevin dynamics (SGLD) based on (Mou et al., 2018; Li et al., 2020) and related approaches (Negrea et al., 2019; Haghifam et al., 2020). In this paper, we unify and substantially generalize stability based generalization bounds and make three technical advances. First, we bound the generalization error of general noisy stochastic iterative algorithms (not necessarily gradient descent) in terms of expected stability, which in turn can be bounded by the expected Le Cam Style Divergence (LSD). Such bounds have a $O(1/n)$ sample dependence unlike many existing bounds with $O(1/\sqrt{n})$ dependence. Second, we introduce Exponential Family Langevin Dynamics (EFLD) which is a substantial generalization of SGLD and which allows exponential family noise to be used with gradient descent. We establish data-dependent expected stability based generalization bounds for general EFLD. Third, we consider an important new special case of EFLD: noisy sign-SGD, which extends sign-SGD by using Bernoulli noise over $\{-1, +1\}$, and we establish optimization guarantees for the algorithm. Further, we present empirical results on benchmark datasets to illustrate the our bounds are non-vacuous and quantitatively much sharper than existing bounds.

## 1 INTRODUCTION

Recent years have seen renewed interest and advances in characterizing generalization performance of learning algorithms in terms of stability, which considers change in performance of a learning algorithm based on change of a single training point (Hardt et al., 2016; Bousquet & Elisseeff, 2002; Li et al., 2020; Mou et al., 2018). For stochastic gradient descent (SGD), Hardt et al. (2016) established generalization bounds based on uniform stability (Hardt et al., 2016; Bousquet & Elisseeff, 2002), although the analysis needed rather small step sizes $\eta_t = 1/t$ which is not useful in practice. While improving the analysis for SGD has remained a challenge, advances have been made on noisy SGD algorithms, especially stochastic gradient Langevin dynamics (SGLD) (Welling & Teh, 2011; Mou et al., 2018; Li et al., 2020), which adds Gaussian noise to the stochastic gradients of marginal variance $\sigma_t^2$. In parallel, there has been key developments on related information-theoretic generalization bounds applicable to SGLD type algorithms (Negrea et al., 2019; Haghifam et al., 2020; Xu & Raginsky, 2017; Russo & Zou, 2016; Pensia et al., 2018).

While these developments have led to major advances in analyzing generalization of noisy SGD algorithms, they each have certain limitations which leave room for further improvements. Using uniform stability, Mou et al. (2018) established a bound for SGLD of the form $\frac{K}{n}\sqrt{\sum_t \eta_t^2/\sigma_t^2}$ which depends on $K$, the global Lipschitz constant for the loss, and with step size $\eta_t \leq \sigma_t \ln 2/K$. The bound has a desirable dependency of $O(1/n)$ on the samples, but has an undesirable dependence on $K$, and the step sizes, bounded by $\sigma_t/K$, are too small. Mou et al. (2018) also presented another bound which addresses some of these issues, but gets an undesirable $O(1/\sqrt{n})$ sample dependence. By building on the developments of Russo & Zou (2016); Xu & Raginsky (2017); Pensia et al. (2018), Negrea et al. (2019) made advances from the information theoretic perspective and established bounds for SGLD which have the desirable dependence on the norm of gradient incoherence, i.e., difference in gradients over different mini-batches, avoids dependence on Lipschitz constant $K$, and is applicable to unbounded sub-Gaussian losses, but have an undesirable $O(1/\sqrt{n})$ sample

dependence. Haghifam et al. (2020) made further advances on the problem from an information theoretic perspective based on the conditional mutual information framework of Steinke & Zakynthinou (2020) and obtained generalization bounds based on gradient incoherence with $O(1/n)$ sample dependence, but their analysis holds for full batch Langevin dynamics, not mini-batch SGLD. Li et al. (2020) made advances on such bounds based on the notion of Bayes-stability, by combining ideas from PAC-Bayes bounds into stability, and established a bound of the form $\frac{c}{n}\sqrt{\sum_t \eta_t^2 \mathbf{g}_e(t)/\sigma_t^2}$ for bounded losses, where $\mathbf{g}_e(t)$ is the expected gradient norm square at step $t$. While the bound avoids dependency on the Lipschitz constant $K$, the dependence on the gradient norm makes such bounds much weaker than the information theoretic bounds of Negrea et al. (2019); Haghifam et al. (2020) which depend on the norm of gradient incoherence, which are typically orders of magnitude smaller. Further, the analysis of Li et al. (2020) still needs small step sizes, bounded by $\sigma_t/K$.

In this paper, we build on the core strengths of such existing approaches, most notably the $O(1/n)$ sample dependence of stability based bounds (Mou et al., 2018; Li et al., 2020) and the dependence on gradient incoherence for information theoretic bounds (Negrea et al., 2019; Haghifam et al., 2020), and develop a framework (Section 2) for developing generalization bounds for noisy stochastic iterative (NSI) algorithms. Our framework considers generalization based on the concept of *expected stability*, rather than uniform stability (Hardt et al., 2016; Bousquet & Elisseeff, 2002; Bousquet et al., 2020; Mou et al., 2018), which yields distribution dependent generalization bounds and avoids the worst-case setting of uniform stability. Building on Li et al. (2020), we show that expected stability of general NSI algorithms can be bounded by the expected Le Cam Style Divergence with dependence on parameter distributions from mini-batches differing by one sample. In Section 3, we introduce Exponential Family Langevin Dynamics (EFLD), a family of noisy gradient descent algorithms based on exponential family noise. Special cases of EFLD include SGLD and noisy versions of Sign-SGD or quantized SGD algorithms widely used in practice (Bernstein et al., 2018a;b; Jin et al., 2020; Alistarh et al., 2017)Our main result provides an expected stability based generalization bound applicable to any EFLD algorithm with a $O(1/n)$ sample dependence and a dependence on gradient incoherence, rather than gradient norms. Existing generalization bounds for SGLD (Li et al., 2020; Negrea et al., 2019) usually use properties of the Gaussian distribution, and our analysis on EFLD illustrates that this was unnecessary. We also consider optimization guarantees for EFLD and establish such results for noisy Sign-SGD and SGLD. Through experiments on benchmark datasets (Section 4), we illustrate that our bounds are non-vacuous and quantitatively much sharper than existing bounds (Li et al., 2020; Negrea et al., 2019).

**Related work.** Uniform stability has been a classical approach for bounding generalization error (Bousquet & Elisseeff, 2002; Bousquet et al., 2020; Feldman & Vondrak, 2018; 2019), pioneered by Rogers & Wagner (1978); Devroye & Wagner (1979). Beyond the aforementioned work, there has been recent work on differential privacy that analyzes the uniform stability of differentially private SGD (DP-SGD) (Hardt et al., 2016; Bassily et al., 2020). Beyond uniform stability, information-theoretic approaches (Russo & Zou, 2016; Xu & Raginsky, 2017) that bounds the generalization error by the mutual information between the algorithm input $S$ and the algorithm output $\mathbf{w}$, have been used for deriving generalization bounds for noisy iterative algorithms (Pensia et al., 2018; Bu et al., 2019). Along this line of literature, Negrea et al. (2019); Haghifam et al. (2020); Rodríguez-Gálvez et al. (2021) prove data-dependent generalization bounds dropping dependence on the Lipschitz constant. Further, tighter bounds (Haghifam et al., 2020; Zhou et al., 2021; Rodríguez-Gálvez et al., 2021; Neu, 2021; Hellström & Durisi, 2021) are proposed based on conditional mutual information (Steinke & Zakynthinou, 2020; Grünwald et al., 2021; Hellström & Durisi, 2020). Due to space limitations, an extended discussion of the related work is deferred to Appendix A.

## 2 GENERALIZATION BOUNDS WITH EXPECTED STABILITY

In the setting of statistical learning, there is an instance space $\mathcal{Z}$, a hypothesis space $\mathcal{W}$, and a loss function $\ell : \mathcal{W} \times \mathcal{Z} \mapsto \mathbb{R}_+$. Let $D$ be an unknown distribution of $\mathcal{Z}$ and let $S \sim D^n$ be $n$ i.i.d. draws from $D$. For any specific hypothesis $\mathbf{w} \in \mathcal{W}$, the population and empirical loss are respectively given by $L_D(\mathbf{w}) \triangleq \mathbb{E}_{z \sim D}[\ell(\mathbf{w}, z)]$, and $L_S(\mathbf{w}) \triangleq \frac{1}{n}\sum_{i=1}^{n}\ell(\mathbf{w}, z_i)$. For any distribution $P$ over the hypothesis space, we respectively denote the expected population and empirical loss as

$$L_D(P) \triangleq \mathbb{E}_{z \sim D}\mathbb{E}_{\mathbf{w} \sim P}[\ell(\mathbf{w}, z)], \qquad \text{and} \qquad L_S(P) \triangleq \frac{1}{n}\sum_{i=1}^{n}\mathbb{E}_{\mathbf{w} \sim P}[\ell(\mathbf{w}, z_i)]. \qquad (1)$$

Consider a randomized algorithm $A$ which works with $S = \{z_1, \ldots, z_n\} \sim D^n$ and cre-

ates a distribution over the hypothesis space $\mathcal{W}$. For convenience, we will denote the distribution as $A(S)$. The focus of the analysis is to bound the generalization error of $A$ defined as: $\text{gen}(A(S)) \triangleq L_D(A(S)) - L_S(A(S))$. We will assume $A$ is permutation invariant, i.e., the ordering of samples in $S$ do not modify $A(S)$, an assumption satisfied by most learning algorithms. We will focus on developing bounds for the expectation $\mathbb{E}_S[L_D(A(S)) - L_S(A(S))]$, and discuss high-probability bounds in the Appendix B.

## 2.1 BOUNDS BASED ON EXPECTED STABILITY

We start our analysis by noting that the expected generalization error can be upper bounded by *expected stability* based on the Hellinger divergence (Sason & Verdu, 2016; Li et al., 2020): $H^2(P\|P') = \frac{1}{2} \int_{\mathbf{w}} (\sqrt{p(\mathbf{w})} - \sqrt{p'(\mathbf{w})})^2 d\mathbf{w}$.

**Proposition 1.** *Let $S_n \sim D^n$ and let $S'_n$ be a dataset obtained by replacing $z_n \in S_n$ with $z'_n \sim D$. Let $A(S_n), A(S'_n)$ respectively denote the distributions over the hypothesis space $\mathcal{W}$ obtained by running randomized algorithm $A$ on $S_n, S'_n$. Assume that for $S_n \sim D^n, \forall z \in \mathcal{Z}$, $\mathbb{E}_{W \sim A(S_n)}[\ell^2(W, z)] \leq c_0/2, c_0 > 0$. With $H(\cdot, \cdot)$ denoting the Hellinger divergence, we have*

$$|\mathbb{E}_{S_n \sim D^n}[L_D(A(S_n)) - L_S(A(S_n))]| \leq c_0 \mathbb{E}_{S_n \sim D^n} \mathbb{E}_{z'_n \sim D} \sqrt{2H^2(A(S_n), A(S'_n))}. \quad (2)$$

**Remark 2.1.** Proposition 1 does not need bounded losses. Just the *second moment* of $\ell(W, z), W \sim A(S_n), S_n \sim D^n, \forall z \in \mathcal{Z}$ need to be bounded. The assumption is satisfied by bounded losses. It is instructive to compare the assumption to that in recent information theoretic bounds (Haghifam et al., 2020; Xu & Raginsky, 2017), where one assumes $\ell(\mathbf{w}, Z), Z \sim D, \forall \mathbf{w} \in \mathcal{W}$ to be *sub-Gaussian*.

**Remark 2.2.** The bound in Proposition 1 is in terms of *expected stability* where we consider $\mathbb{E}_{S \sim D^n} \mathbb{E}_{z'_n \sim D}[\cdots]$, an important departure from bounds based on *uniform stability* (Elisseeff et al., 2005; Bousquet & Elisseeff, 2002; Mou et al., 2018; Bousquet et al., 2020; Feldman & Vondrak, 2018; 2019) where one considers $\sup_{S,S' \in \mathcal{Z}^n, |S \setminus S'|=1}[\cdots]$. Replacing sup by $\mathbb{E}$ makes the bounds distribution dependent, and arguably leads to quantitatively tighter bounds.

Note that the Hellinger divergence can be bounded by the KL divergence.

**Proposition 2.** *For any distributions $P$ and $P'$, $2H^2(P, P') \leq \min\{KL(P, P'), \sqrt{\frac{1}{2}KL(P, P')}\}$.*

## 2.2 EXPECTED STABILITY OF NOISY STOCHASTIC ITERATIVE ALGORITHMS

We consider a general family of noisy stochastic iterative (NSI) algorithms. Given $S \sim D^n$, such iterative algorithms have two additional sources of randomness in each iteration $t$: (a) a stochastic mini-batch of samples $S_{B_t}$, with $|S_{B_t}| = b$, drawn uniformly at random with replacement from $S$; and (b) noise $\boldsymbol{\xi}_t$ suitably included in the iterative update. Given a trajectory of past iterates $W_{0:(t-1)} = \mathbf{w}_{0:(t-1)}$, the new iterate $W_t$ is drawn from a distribution $P_{B_t, \boldsymbol{\xi}_t | \mathbf{w}_{0:(t-1)}}$ over $\mathcal{W}$:

$$W_t \sim P_{B_t, \boldsymbol{\xi}_t | \mathbf{w}_{0:(t-1)}}(W). \quad (3)$$

We will drop the conditioning $\mathbf{w}_{0:(t-1)}$ to avoid clutter in the sequel. Let $P_T, P'_T$ denote the marginal distributions over hypotheses $\mathbf{w} \in \mathcal{W}$ after $T$ steps of the algorithm based on $S_n, S'_n$ respectively. Further, let $P_{0:(t-1)}$ denote the joint distribution over $W_{0:(t-1)} = (W_0, \ldots, W_{t-1})$, and let $P_{t|} \equiv P_{B_t, \boldsymbol{\xi}_t | \mathbf{w}_{0:(t-1)}}$ compactly denote the conditional distribution on $W_t$ conditioned on the trajectory $W_{0:(t-1)} = \mathbf{w}_{0:t-1}$. Following (Negrea et al., 2019; Haghifam et al., 2020; Pensia et al., 2018), we use the following chain rule for KL-divergence: $KL(P_T\|P'_T) \leq KL(P_{0:T}\|P'_{0:T}) = \sum_{t=1}^{T} \mathbb{E}_{P_{0:(t-1)}}[KL(P_{t|}\|P'_{t|})]$. Let $\bar{S} \sim D^{n+1}$, and let $S_n, S'_n$ be size $n$ subsets of $\bar{S}$ such that $S_n = \{Z_1, \ldots, Z_{n-1}, Z_n\}$ and $S'_n = \{Z_1, \ldots, Z_{n-1}, Z'_n\}$, where $Z'_n = Z_{n+1}$. Let $S_0 = \{Z_1, \ldots, Z_{n-1}\}$. The algorithms we consider use a mini-batch of size $b$ in each iteration uniformly sampled from $n$ samples. Let the set of all mini-batch index sets be denoted by $G$. Let the set of all mini-batch index sets $A$ drawn from $S_0$ be denoted by $G_0$. Note that $|G_0| = \binom{n-1}{b}$. Let $G_1$ denote the set of all mini-batch index sets $B$ which includes the last sample, viz. $z_n$ for $S$ with mini-batches and $z'_n$ for $S'_n$. Note that $|G_1| = \binom{n-1}{b-1}$. Also note that $|G_0| + |G_1| = \binom{n-1}{b} + \binom{n-1}{b-1} = \binom{n}{b} = |G|$.

Following Li et al. (2020), we can bound their conditional KL-divergences $KL(P_{t|}\|P'_{t|})$ in terms of a Le Cam Style Divergence (LSD). While the classical Le Cam divergence (Sason & Verdu,

2016) is $LCD(P\|P') \triangleq \frac{1}{2}\int \frac{(dP-dP')^2}{dP+dP'}$ (where $dP$ denotes the density), our bounds in terms of $LSD(P_{t|}\|\|P'_{t|}) \triangleq \int \frac{(dP_{B_t,\boldsymbol{\xi}_t}-dP'_{B_t,\boldsymbol{\xi}_t})^2}{dP_{A_t,\boldsymbol{\xi}_t}}$, $B_t \in G_1, A_t \in G_0$. Note that $P_{B_t,\boldsymbol{\xi}_t}$ and $P'_{B_t,\boldsymbol{\xi}_t}$ represent the distribution of $W_t$ for $S_n$ and $S'_n$ respectively since the mini-batch $S_{B_t}$ of $S_n$ and $S'_n$ differs in the $n$-th sample. Putting everything together, we have the following LSD based bound.

**Lemma 1.** *Consider a noisy stochastic iterative algorithms of the form* (3) *with mini-batch size* $b \le n/2$. *Then, with* $c_1 = \sqrt{2}c_0$ *(with* $c_0$ *as in Proposition 1), we have*

$$|\mathbb{E}_{S_n}[L_D(A(S_n))-L_{S_n}(A(S_n))]| \le c_1 \frac{b}{n}\mathbb{E}_{S_n}\mathbb{E}_{z'_n}\sqrt{\sum_{t=1}^{T}\mathop{\mathbb{E}}_{W_{0:(t-1)}}\mathop{\mathbb{E}}_{B_t \in G_1}\mathop{\mathbb{E}}_{A_t \in G_0}\left[\int_{\boldsymbol{\xi}_t}\frac{\left(dP_{B_t,\boldsymbol{\xi}_t}-dP'_{B_t,\boldsymbol{\xi}_t}\right)^2}{dP_{A_t,\boldsymbol{\xi}_t}}d\boldsymbol{\xi}_t\right]}.$$

(4)

**Remark 2.3.** Li et al. (2020) essentially has this result for SGLD and inspired our work. Our proofs are significantly simpler and, more importantly, illustrates applicability to general noisy iterative algorithms of the form (3), not just SGLD with Gaussian noise as in Li et al. (2020).

**Remark 2.4.** Note that the bound does not assume the loss to be bounded, depends on expectations over samples $S_n, z'_n$, trajectories $\mathbf{w}_{0:(t-1)}$, and mini-batches $B_t, A_t$. Further, the bound depends on the distribution discrepancy as captured by the expected LSD.

**Remark 2.5.** The bound seems to worsen with $b$, the size of the mini-batch. As we shown in Section 3, the expected LSD term has a $\frac{1}{b^2}$ dependence for the Exponential Family Langevin dynamics (EFLD) models we introduce, so the leading $b$ is neutralized.

## 3 EXPONENTIAL FAMILY LANGEVIN DYNAMICS

In recent years, considerable advances have been made in establishing generalization bounds for stochastic gradient Langevin dynamics (SGLD) (Li et al., 2020; Pensia et al., 2018; Negrea et al., 2019; Haghifam et al., 2020). As an example of NSI algorithms of the form (3), SGLD adds an isotropic Gaussian noise at every step of SGD: $\mathbf{w}_{t+1} = \mathbf{w}_t - \eta_t\nabla\ell(\mathbf{w}_t, S_{B_t}) + \mathcal{N}\left(0, \sigma_t^2\mathbb{I}_d\right)$, where $\nabla\ell(\mathbf{w}_t, S_{B_t})$ is the stochastic gradient on mini-batch $B_t$, $\eta_t$ is the step size, and $\sigma_t^2$ is noise vairance.

In this paper, we introduce a substantial generalization of SGLD called Exponential Family Langevin Dynamics (EFLD) which uses general exponential family noise in noisy iterative updates of the form (3). In addition to being a mathematical generalization of the popular SGLD, the proposed EFLD provides flexibility to use noise gradient algorithms with different representation of the gradient, e.g., Bernoulli noise for Sign-SGD, discrete distribution for quantized or finite precision SGD, etc. (Canonne et al., 2020; Alistarh et al., 2017; Jiang & Agrawal, 2018; Yang et al., 2019).

### 3.1 EXPONENTIAL FAMILY LANGEVIN DYNAMICS (EFLD)

Exponential families (Barndorff-Nielsen, 2014; Brown, 1986; Wainwright & Jordan, 2008) constitute a large family of parametric distributions which include Gaussian, Bernoulli, gamma, Poisson, Dirichlet, etc., as special cases. Exponential families are typically represented in terms of natural parameters $\boldsymbol{\theta}$, and we consider component-wise independent distributions with scaled natural parameter $\boldsymbol{\theta}_\alpha = \boldsymbol{\theta}/\alpha$ with scaling $\alpha > 0$, i.e., $p_\psi(\boldsymbol{\xi}, \boldsymbol{\theta}_\alpha) = \exp(\langle\boldsymbol{\xi}, \boldsymbol{\theta}_\alpha\rangle - \psi(\boldsymbol{\theta}_\alpha))\boldsymbol{\pi}_0(\boldsymbol{\xi}) = \prod_{j=1}^p \exp(\xi_j\theta_{j\alpha} - \psi_j(\theta_{j\alpha}))\pi_0(\xi_j)$, where $\boldsymbol{\xi}$ is the sufficient statistic, $\psi(\boldsymbol{\theta}_\alpha) = \sum_{j=1}^p \psi_j(\theta_{j\alpha})$ is the log-partition function, and $\boldsymbol{\pi}_0(\xi) = \prod_{j=1}^p \pi_0(\xi_j)$ is the base measure. Note that $\alpha = 1$ gives the canonical form of the exponential family distributions. For general scaling $\alpha > 0$, for some cases the base measure $\pi_0$ may depend on the scaling, i.e., $\pi_{0,\alpha}$. A scaling $\alpha > 0$ is valid as long as $\exp(\langle\boldsymbol{\xi}, \boldsymbol{\theta}_\alpha\rangle$ is integrable, i.e., $\int_{\boldsymbol{\xi}}\exp(\langle\boldsymbol{\xi}, \boldsymbol{\theta}_\alpha\rangle\boldsymbol{\pi}_0(\boldsymbol{\xi})d\boldsymbol{\xi} < \infty$. Further, $\psi$ is a smooth function by construction (Barndorff-Nielsen, 2014; Banerjee et al., 2005; Wainwright & Jordan, 2008) and the smoothness of $\psi$ implies $\nabla^2\psi(\boldsymbol{\theta}_\alpha) \le c_2\mathbb{I}$.

Exponential family Langevin dynamics (EFLD) uses noisy stochastic gradient updates similar to SGLD, but using exponential family noise rather than Gaussian noise as in SGLD. In particular, for mini-batch $S_{B_t}$, EFLD updates are as follows: with step size $\rho_t > 0$

$$\mathbf{w}_t = \mathbf{w}_{t-1} - \rho_t\boldsymbol{\xi}_t, \qquad \boldsymbol{\xi}_t \sim p_\psi(\boldsymbol{\xi}; \boldsymbol{\theta}_{B_t,\alpha_t}),$$

(5)

where

$$p_\psi(\boldsymbol{\xi}; \boldsymbol{\theta}_{B_t,\alpha_t}) = \exp(\langle\boldsymbol{\xi}, \boldsymbol{\theta}_{B_t,\alpha_t}\rangle - \psi(\boldsymbol{\theta}_{B_t,\alpha_t}))\boldsymbol{\pi}_0(\boldsymbol{\xi}), \qquad \boldsymbol{\theta}_{B_t,\alpha_t} \triangleq \frac{\boldsymbol{\theta}_{B_t}}{\alpha_t} = \frac{\nabla\ell(\mathbf{w}_{t-1}, S_{B_t})}{\alpha_t}.$$

(6)

For EFLD, the natural parameter $\boldsymbol{\theta}_{B_t,\alpha_t}$ at step $t$ is simply a scaled version of the mini-batch gradient $\nabla\ell(\mathbf{w}_{t-1}, S_{B_t})$. We first show that EFLD becomes SGLD when the exponential family is Gaussian, and becomes a noisy version of sign-SGD (Bernstein et al., 2018a;b) when the exponential family is Bernoulli over $\{-1, +1\}$. More details and examples are in Appendix C.1.

**Example 3.1** (SGLD). SGLD uses scaled Gaussian noise with $\psi(\boldsymbol{\theta}) = \|\boldsymbol{\theta}\|_2^2/2, \alpha_t = \sqrt{\sigma_t/\eta_t}$, so that $p_\psi(\boldsymbol{\xi}; \boldsymbol{\theta}_{B_t,\alpha_t}) = \mathcal{N}(\boldsymbol{\theta}_{B_t}, \alpha_t^2\mathbb{I}_d)$. By taking $\rho_t = \sqrt{\eta_t\sigma_t}$, the update (5) based on $\rho_t\boldsymbol{\xi}_t$ is distributed as $\mathcal{N}(\rho_t\boldsymbol{\theta}_{B_t}, \rho_t^2\alpha_t^2\mathbb{I}_d) = \mathcal{N}(\eta_t\nabla\ell(\mathbf{w}_{t-1}, S_{B_t}), \sigma_t^2\mathbb{I}_d)$. Thus the EFLD update reduces to the SGLD update: $\mathbf{w}_t = \mathbf{w}_{t-1} - \eta_t\nabla\ell(\mathbf{w}_{t-1}, S_{B_t}) + \mathcal{N}\left(0, \sigma_t^2\mathbb{I}_d\right)$. □

**Example 3.2** (Noisy Sign-SGD). By taking $\rho_t = \eta_t$ and component-wise $\xi_j \in \{-1, 1\}$, $\pi_0(\xi_j) = 1$, $\psi(\theta) = \log(\exp(-\theta) + \exp(\theta))$ in exponential family update equation (5), the $j$-th component of exponential family distribution $p_\psi(\boldsymbol{\xi}; \boldsymbol{\theta}_{B_t,\alpha_t})$ becomes $p_{\boldsymbol{\theta}_{B_t,\alpha_t,j}}(\xi_j) = \frac{\exp(\xi_j\boldsymbol{\theta}_{B_t,\alpha_t,j})}{\exp(-\boldsymbol{\theta}_{B_t,\alpha_t,j}) + \exp(\boldsymbol{\theta}_{B_t,\alpha_t,j})}$. Thus, the EFLD update reduces to a noisy version of Sign-SGD: $\mathbf{w}_t = \mathbf{w}_{t-1} - \eta_t\boldsymbol{\xi}_t$, $\boldsymbol{\xi}_{t,j} \sim p_{\boldsymbol{\theta}_{B_t,\alpha_t,j}}(\xi_j), j \in [d]$, where $\boldsymbol{\theta}_{B_t,\alpha_t} = \nabla\ell(\mathbf{w}_{t-1}, S_{B_t})/\alpha_t$ is the scaled mini-batch gradient. □

### 3.2 EXPECTED STABILITY OF EXPONENTIAL FAMILY LANGEVIN DYNAMICS

From Lemma 1, conditioned on a trajectory $\mathbf{w}_{0:(t-1)}$, mini-batches $S_{B_t}, S_{A_t}$, we can get generalization bound by suitably bounding the Le Cam Style Divergence (LSD) given by: $I_{A_t,B_t} = \int_{\boldsymbol{\xi}_t} \frac{(dP_{B_t,\boldsymbol{\xi}_t} - dP'_{B_t,\boldsymbol{\xi}_t})^2}{dP_{A_t,\boldsymbol{\xi}_t}} d\boldsymbol{\xi}_t$. For EFLD, the density functions $dP_{B_t,\boldsymbol{\xi}_t}$ are exponential family densities $p_\psi(\boldsymbol{\xi}; \boldsymbol{\theta}_{B_t,\alpha_t})$ as in (5)-(6), and we have the following bound on the per step LSD:

**Theorem 1.** *For a given set* $\bar{S} \sim D^{n+1}$ *and* $\mathbf{w}_{t-1}$ *at iteration* $(t-1)$, *let* $\Delta_{t|\mathbf{w}_{t-1}}(\bar{S}) = \max_{z,z'\in\bar{S}} \|\nabla\ell(\mathbf{w}_{t-1}, z) - \nabla\ell(\mathbf{w}_{t-1}, z')\|_2$. *Further, for a* $c_2$-*smooth log-partition function* $\psi$, *let the scaling* $\alpha_{t|\mathbf{w}_{t-1}}$ *be data-dependent such that* $\alpha_{t|\mathbf{w}_{t-1}}^2 \geq 8c_2\Delta_{t|\mathbf{w}_{t-1}}^2(S_{n+1})$. *Then, we have*

$$I_{A_t,B_t} \leq 5c_2\|\boldsymbol{\theta}_{B_t,\alpha_t} - \boldsymbol{\theta}_{B'_t,\alpha_t}\|_2^2 = \frac{5c_2}{2\alpha_{t|\mathbf{w}_{t-1}}^2}\left[\|\nabla\ell(\mathbf{w}_{t-1}, S_{B_t}) - \nabla\ell(\mathbf{w}_{t-1}, S'_{B_t})\|_2^2\right], \quad (7)$$

Note that $S_{B_t}$ and $S'_{B_t}$ only differ at samples $z_n$ and $z'_n$. The above bound can now be directly applied to Lemma 1 to get expected stability based generalization bounds for any EFLD algorithm.

**Theorem 2.** *Consider an exponential family Langevin dynamics (EFLD) algorithm of the form* (5)-(6) *with a* $c_2$-*smooth log-partition function* $\psi$. *Then, for mini-batch size* $b \leq n/2$, *with* $c = c_0\sqrt{5c_2}$ *(with* $c_0$ *as in Lemma 1) and* $\alpha_{t|}^2 \geq 8c_2\Delta_{t|}^2(S_{n+1})$ *(as in Theorem 1, with the conditioning on* $\mathbf{w}_{t-1}$ *hidden to avoid clutter), we have*

$$|\mathbb{E}_S[L_D(A(S)) - L_S(A(S))]| \leq c\frac{1}{n}\mathop{\mathbb{E}}_{S_{n+1}}\sqrt{\sum_{t=1}^T \mathop{\mathbb{E}}_{W_{0:(t-1)}}\frac{1}{\alpha_{t|}^2}\left[\|\nabla\ell(\mathbf{w}_{t-1}, z_n) - \nabla\ell(\mathbf{w}_{t-1}, z'_n)\|_2^2\right]}.$$
$$(8)$$

**Remark 3.1.** Theorem 2 captures the generalization error of SGLD, which is a special case of EFLD. Our bound has the same dependence on $n$, $T$, step size $\eta_t$ as the bound in Li et al. (2020). However, our bound is numerically sharper because we replace the *gradient norms*, i.e., $\frac{1}{n}\sum_{z\in S}\|\ell(\mathbf{w}_t, z)\|$ in Li et al. (2020) and with gradient discrepancy $\|\nabla\ell(\mathbf{w}_t, z) - \nabla\ell(\mathbf{w}_t, z')\|$, which is quantitatively smaller than gradient norms as we show in the experiment section. The bound in Negrea et al. (2019) depends on *gradient incoherence* which is empirically smaller than gradient discrepancy as observed in the experiment section, their bound depends on $1/\sqrt{n}$, which is worse than the $1/n$ dependence in our bound.

**Remark 3.2.** EFLD can be extended to work with anisotropic noise by using $\boldsymbol{\theta}_{B_t,\boldsymbol{\alpha}_t} = \nabla\ell(\mathbf{w}_{t-1}, S_{B_t}) \oslash \boldsymbol{\alpha}_t$ in (6) where $\boldsymbol{\alpha}_t \in \mathbb{R}^p$ determines different scaling for each dimension and $\oslash$ denotes Hadamard division. Theorems 1 and 2 can be extended to such anisotropic noise by using $\boldsymbol{\alpha}$-scaled norms for the gradient discrepancy, i.e., $\|\mathbf{g} - \mathbf{g}'\|_{2,\boldsymbol{\alpha}}^2 = \sum_j (g_j - g'_j)^2/\alpha_j^2$. □

**Remark 3.3.** The condition on $\alpha_t$ is a data-dependent quantity, which can be computed along the training process. It gives much more benign condition of the step size compared to those in the related work (Mou et al., 2018; Li et al., 2020, Hardt et al. 2016), which require step size being bounded by $\sigma_t/L$. However, the step sizes in Theorem 2 need to be bounded by $\sigma_t/\Delta_t(\bar{S})$, which is considerably more relaxed since $\Delta_t(\bar{S})$ is much smaller than Lipschitz constant $L$, which is a uniform bound over the whole parameter space. Also, usually one would expect $\Delta_t(\bar{S})$ to decrease as training proceeds since the gradients shrink as the loss function being minimized. Thus, the constraint on step size does not require the step sizes to be as small as $\sigma_t/L$.

### 3.3 Proof Sketches of Main Results: Theorems 1 and 2

We focus on Theorem 1. To avoid clutter, we drop the subscript $t$ for the analysis and note that the analysis holds for any step $t$. When the density $dP_{B,\boldsymbol{\xi}} = p_\psi(\boldsymbol{\xi}; \boldsymbol{\theta}_{B,\alpha})$, by mean-value theorem, for each $\boldsymbol{\xi}$, we have $p_\psi(\boldsymbol{\xi}; \boldsymbol{\theta}_{B,\alpha}) - p_\psi(\boldsymbol{\xi}; \boldsymbol{\theta}_{B',\alpha}) = \langle \boldsymbol{\theta}_{B,\alpha} - \boldsymbol{\theta}_{B,\alpha}, \nabla_{\tilde{\boldsymbol{\theta}}_{B,\alpha}} p_\psi(\boldsymbol{\xi}; \tilde{\boldsymbol{\theta}}_{B,\alpha}) \rangle$, for some $\tilde{\boldsymbol{\theta}}_{B,\alpha} = \gamma_{\boldsymbol{\xi}} \boldsymbol{\theta}_{B,\alpha} + (1 - \gamma_{\boldsymbol{\xi}}) \boldsymbol{\theta}'_{B,\alpha}$ where $\gamma_{\boldsymbol{\xi}} \in [0, 1]$. Then,

$$I_{A,B} = \int_{\boldsymbol{\xi}} \frac{\left(p_\psi(\boldsymbol{\xi}; \boldsymbol{\theta}_{B,\alpha}) - p_\psi(\boldsymbol{\xi}; \boldsymbol{\theta}_{B',\alpha})\right)^2}{p_\psi(\boldsymbol{\xi}; \boldsymbol{\theta}_{A,\alpha})} d\boldsymbol{\xi} = \int_{\boldsymbol{\xi}} \frac{\langle \boldsymbol{\theta}_{B,\alpha} - \boldsymbol{\theta}'_{B,\alpha}, \boldsymbol{\xi} - \nabla_{\tilde{\boldsymbol{\theta}}_{B,\alpha}} \psi(\boldsymbol{\xi}; \tilde{\boldsymbol{\theta}}_{B,\alpha}) \rangle^2 \, p_\psi^2(\boldsymbol{\xi}; \tilde{\boldsymbol{\theta}}_{B,\alpha})}{p_\psi(\boldsymbol{\xi}; \boldsymbol{\theta}_{A,\alpha})} d\boldsymbol{\xi} \,,$$

since $p_\psi(\boldsymbol{\xi}; \tilde{\boldsymbol{\theta}}_{B,\alpha}) = \exp(\langle \boldsymbol{\xi}, \tilde{\boldsymbol{\theta}}_{B,\alpha} \rangle - \psi(\tilde{\boldsymbol{\theta}}_{B,\alpha})) \pi_0(\boldsymbol{\xi})$.

**Handling Distributional Dependence of $\tilde{\boldsymbol{\theta}}_B$.** Note that we cannot proceed with the analysis with the density term depending on $\tilde{\boldsymbol{\theta}}_{B,\alpha}$ since $\tilde{\boldsymbol{\theta}}_{B,\alpha}$ depends on $\boldsymbol{\xi}$. So, we first bound the density term depending on $\tilde{\boldsymbol{\theta}}_{B,\alpha}$ in terms of exponential family densities with parameters $\boldsymbol{\theta}_{B,\alpha}$ and $\boldsymbol{\theta}_{B,\alpha}$ using $c_2$-smoothness of $\psi$.

**Lemma 2.** *For some $\gamma_{\boldsymbol{\xi}} \in [0, 1]$, $\tilde{\boldsymbol{\theta}}_{B,\alpha} = \gamma_{\boldsymbol{\xi}} \boldsymbol{\theta}_{B,\alpha} + (1 - \gamma_{\boldsymbol{\xi}}) \boldsymbol{\theta}'_{B,\alpha}$, we have*

$$\frac{\exp\left[\langle \boldsymbol{\xi}, \tilde{\boldsymbol{\theta}}_{B,\alpha} \rangle - \psi(\tilde{\boldsymbol{\theta}}_{B,\alpha})\right]}{\max\left(\exp\left[\langle \boldsymbol{\xi}, \boldsymbol{\theta}_{B,\alpha} \rangle - \psi(\boldsymbol{\theta}_{B,\alpha})\right], \exp\left[\langle \boldsymbol{\xi}, \boldsymbol{\theta}_{B',\alpha} \rangle - \psi(\boldsymbol{\theta}_{B',\alpha})\right]\right)} \le \exp\left[c_2 \|\boldsymbol{\theta}_{B,\alpha} - \boldsymbol{\theta}_{B',\alpha}\|_2^2\right] \,.$$

**Bounding the Density Ratio.** Next we focus on the density ratio $p_\psi^2(\boldsymbol{\xi}, \tilde{\boldsymbol{\theta}}_{B,\alpha})/p_\psi(\boldsymbol{\xi}; \boldsymbol{\theta}_{A,\alpha})$. By Lemma 2, it suffices to focus on $p_\psi^2(\boldsymbol{\xi}, \boldsymbol{\theta}_{B,\alpha})/p_\psi(\boldsymbol{\xi}; \boldsymbol{\theta}_{A,\alpha})$ or the equivalent term for $\boldsymbol{\theta}_{B',\alpha}$. We show that the density ratio can be bounded by another exponential family with parameters $(2\boldsymbol{\theta}_{B,\alpha} - \boldsymbol{\theta}_{A,\alpha})$.

**Lemma 3.** *For any $\boldsymbol{\xi}$, we have*

$$\frac{\exp\left[\langle \boldsymbol{\xi}, 2\boldsymbol{\theta}_{B,\alpha} \rangle - 2\psi(\boldsymbol{\theta}_{B,\alpha})\right]}{\exp\left[\langle \boldsymbol{\xi}, \boldsymbol{\theta}_{A,\alpha} \rangle - \psi(\boldsymbol{\theta}_{A,\alpha})\right]} \le \exp\left[2c_2 \|\boldsymbol{\theta}_{B,\alpha} - \boldsymbol{\theta}_{A,\alpha}\|_2^2\right] \exp\left[\langle \boldsymbol{\xi}, (2\boldsymbol{\theta}_{B,\alpha} - \boldsymbol{\theta}_{A,\alpha}) - \psi(2\boldsymbol{\theta}_{B,\alpha} - \boldsymbol{\theta}_{A,\alpha})\right].$$

The analysis for the term $p_\psi^2(\boldsymbol{\xi}, \boldsymbol{\theta}_{B',\alpha})/p_\psi(\boldsymbol{\xi}; \boldsymbol{\theta}_{A,\alpha})$ is exactly the same.

**Bounding the Integral.** Ignoring multiplicative terms which do not depend on $\boldsymbol{\xi}$ for the moment, the analysis needs to bound an integral term of the form $\int_{\boldsymbol{\xi}} \langle \boldsymbol{\theta}_{B,\alpha} - \boldsymbol{\theta}'_{B,\alpha}, \boldsymbol{\xi} - \nabla\psi(\boldsymbol{\xi}; \tilde{\boldsymbol{\theta}}_{B,\alpha}) \rangle^2 \, p_\psi(\boldsymbol{\xi}; 2\boldsymbol{\theta}_{B,\alpha} - \boldsymbol{\theta}_{A,\alpha}) d\boldsymbol{\xi}$, and a similar term with $p_\psi^2(\boldsymbol{\xi}; 2\boldsymbol{\theta}_{B',\alpha} - \boldsymbol{\theta}_{A,\alpha})$. First, note that $\nabla\psi(\boldsymbol{\xi}; \tilde{\boldsymbol{\theta}}_{B,\alpha}) = \tilde{\boldsymbol{\mu}}_{B,\alpha}$, the expectation parameter for $p_\psi(\boldsymbol{\xi}; \tilde{\boldsymbol{\theta}}_{B,\alpha})$ Wainwright & Jordan (2008); Banerjee et al. (2005). The integral, however, is with respect to $p_\psi(\boldsymbol{\xi}; 2\boldsymbol{\theta}_{B,\alpha} - \boldsymbol{\theta}_{A,\alpha})$. We handle this discrepancy by using $\boldsymbol{\xi} - \nabla\psi(\boldsymbol{\xi}; \tilde{\boldsymbol{\theta}}_{B,\alpha}) = (\boldsymbol{\xi} - \mathbb{E}[\boldsymbol{\xi}]) + (\mathbb{E}[\boldsymbol{\xi}] - \nabla\psi(\boldsymbol{\xi}; \tilde{\boldsymbol{\theta}}_{B,\alpha}))$, and decomposing as sum-of-squares. Quadratic form for the first term yields the covariance $\mathbb{E}[(\boldsymbol{\xi} - \mathbb{E}[\boldsymbol{\xi}])(\boldsymbol{\xi} - \mathbb{E}[\boldsymbol{\xi}])^T] = \nabla^2 \psi(\boldsymbol{\theta}_{2\boldsymbol{\theta}_{B,\alpha} - \boldsymbol{\theta}_{A,\alpha}}) \le c_2 \mathbb{I}$, by smoothness. The second term depends on the difference of gradients $\nabla\psi(2\boldsymbol{\theta}_{B,\alpha} - \boldsymbol{\theta}_{A,\alpha}) - \nabla\psi(\tilde{\boldsymbol{\theta}}_{B,\alpha})$ which, using smoothness and additional analysis, can be bounded by the norm of $(\boldsymbol{\theta}_{B,\alpha} - \boldsymbol{\theta}_{A,\alpha})$. All the pieces can be put together to get the bound in Theorem 1, which when used in Lemma 1 yields Theorem 2.

### 3.4 Optimization Guarantees for EFLD

We now establish optimization guarantees for two examples of EFLD, i.e., Noisy Sign-SGD with Bernoulli noise over $\{-1, +1\}$ and SGLD with Gaussian noise.

**Noisy Sign-SGD.** For mini-batch $B_t$ and scaling $\alpha_t$, mini-batch Noisy Sign-SGD updates the parameters as $\mathbf{w}_t = \mathbf{w}_{t-1} - \eta_t \boldsymbol{\xi}_t$, where each component $j \in [d]$

$$\boldsymbol{\xi}_{t,j} \sim p_{\boldsymbol{\theta}_{B_t, \alpha_t, j}}(x) = \frac{\exp(x\boldsymbol{\theta}_{B_t, \alpha_t, j})}{\exp(-\boldsymbol{\theta}_{B_t, \alpha_t, j}) + \exp(\boldsymbol{\theta}_{B_t, \alpha_t, j})}, \; x \in \{-1, +1\} \tag{9}$$

where $\boldsymbol{\theta}_{B_t, \alpha_t} = \nabla\ell(\mathbf{w}_{t-1}, S_{B_t})/\alpha_t$ is the scaled mini-batch gradient. The full-batch version uses parameters $\mathbb{E}_{B_t}[\boldsymbol{\theta}_{B_t, \alpha_t}] = \nabla L_S(\mathbf{w}_{t-1})$ For the optimization analysis, we assume that the loss is smooth and mini-batch gradients are unbiased, symmetric, and sub-Gaussian.

**Assumption 1.** *The loss function $L_S$ satisfies: for all $\mathbf{w}$ and $\mathbf{w}'$, for some non-negative constant $\vec{K} := [K_1, \ldots, K_d]$, we have $L_S(\mathbf{w}) \leq L_S(\mathbf{w}') + \nabla L_S(\mathbf{w}')^T(\mathbf{w} - \mathbf{w}') + \frac{1}{2}\sum_i K_i(\mathbf{w}_i - \mathbf{w}'_i)^2$.*

**Assumption 2.** *Given $\mathbf{w}_{t-1}$, the mini-batch gradient $\nabla \ell(\mathbf{w}_{t-1}, S_{B_t})$ is (a) unbiased, i.e., $\mathbb{E}_{B_t|\mathbf{w}_{t-1}}\nabla \ell(\mathbf{w}_{t-1}, S_{B_t}) = \nabla L_S(\mathbf{w}_{t-1})$; (b) symmetric, i.e., the density $p(x)$ of $x \equiv \nabla \ell(\mathbf{w}_{t-1}, S_{B_t})$ is symmetric around its expectation $L_S(\mathbf{w}_{t-1})$: $p(x) = p(2\nabla L_S(\mathbf{w}_{t-1}) - x)$ and (c) sub-Gaussian, i.e., for any $\lambda > 0$, any $\mathbf{v}$ s.t. $\|\mathbf{v}\|_2 = 1$, $\mathbb{E}_{B_t|\mathbf{w}_{t-1}} \exp \lambda\langle \mathbf{v}, \nabla \ell(\mathbf{w}_{t-1}, S_{B_t}) - \nabla L_S(\mathbf{w}_{t-1})\rangle \leq \exp(\lambda^2 \kappa_t^2/2)$ for some constant $\kappa_t > 0$.*

Based on the assumptions, we have the following optimization guarantee for mini-batch noisy Sign-SGD. We defer the optimization guarantee for full-batch noisy Sign-SGD to Appendix D.

**Theorem 3.** *Under Assumption 1 and 2, for mini-batch noisy Sign-SGD with step size $\eta_t = 1/\sqrt{T}$, $\alpha_t$ satisfying $c \geq \alpha_t \geq \max[\sqrt{2}\kappa_t, 4\|\nabla L_S(\mathbf{w}_{t-1})\|_\infty]$, we have for any $S$ and any initialization $\mathbf{w}_0$*

$$\mathbb{E}\left[\frac{1}{T}\sum_{t=1}^{T}\|\nabla L_S(\mathbf{w}_t)\|_2^2\right] \leq \frac{4c}{\sqrt{T}}\left(L_S(\mathbf{w}_0) - L_S(\mathbf{w}^*) + \frac{1}{2}\|\vec{K}\|_1\right), \qquad (10)$$

*where the expectation is taken over the randomness of algorithm.*

**SGLD.** We acknowledge that the following optimization result of SGLD exists in various forms, as noisy gradient descent algorithms have been studied in literature such as differential privacy, where SGLD can be viewed as DP-SGD (Bassily et al., 2014; Wang & Xu, 2019) and the proof technique boils down to bounding the stochastic variance of the noisy gradient (Shamir & Zhang, 2013).

**Theorem 4.** *Under Assumption 1 and 2, with $K_i = K, \forall i \in [d]$, for any $S$, SGLD (EFLD with step size $\rho_t = \sqrt{\eta_t \sigma_t}$, $\alpha_t = \sqrt{\sigma_t/\eta_t}$), $|B_t| = b$, and $\eta_t = \frac{1}{\sqrt{T}}$, can achieve*

$$\frac{1}{T}\sum_{t=1}^{T}\mathbb{E}\|\nabla L_S(\mathbf{w}_t)\|^2 \leq O\left(\frac{1}{\sqrt{T}}\right) + O\left(K\frac{p\sum_{t=1}^{T}\alpha_t^4 + \log T}{\sqrt{T}}\right), \qquad (11)$$

*where the expectation is over the randomness of the algorithm.*

The error rate of SGLD depends on the noise variance $\alpha_t$. One can choose a decaying noise variance such as $\alpha_t = 1/\sqrt[4]{t}$ to guarantee the convergence. Then the rate will become $O(\log T/\sqrt{T})$. We note that similar to the optimization guarantees of DP-SGD, the convergence rate depends on the dimension of the gradient $p$ due to the isotropic Gaussian noise. Special noise structures such as anisotropic noise that aligned with the gradient structure can reduce the dependence on dimension (Kairouz et al., 2020; Zhang et al., 2021; Asi et al., 2021; Zhou et al., 2020).

## 4 EXPERIMENTS

In this section, we conduct a series of experiments to evaluate our generalization error bounds. For SGLD, we aim to compare the proposed bound in Theorem 2 with existing bounds in Li et al. (2020), Negrea et al. (2019), and Rodríguez-Gálvez et al. (2021) for various datasets. Note that the bound presented in Rodríguez-Gálvez et al. (2021) is an extension of that in Haghifam et al. (2020) from full-batch setting to mini-batch setting . We also evaluate the optimization performance of proposed Noisy Sign-SGD by comparing it with the original sign-SGD (Bernstein et al., 2018a) and present the corresponding generalization bound in Theorem 2.

The details of our model architectures, learning rate scheduling, hyper-parameter selections and additional experimental results can be found in Appendix E. We acknowledge that we did not achieve the state-of-the art predictive performance, mainly due to the simplicity of our model architectures. With more complex model and further tuning, the prediction results could be improved.

### 4.1 STOCHASTIC GRADIENT LANGEVIN DYNAMICS

**Comparison with existing work.** We have derived theoretical generalization error bounds that depend on the data-dependent quantity *gradient discrepancy*, i.e., $\|\nabla \ell(\mathbf{w}_t, z_n) - \nabla \ell(\mathbf{w}_t, z'_n)\|_2^2$. Existing bounds in Li et al. (2020) and Negrea et al. (2019) have also improved the Lipschitz constant in Mou et al. (2018) to a data-dependent quantity. As shown in Figure 1 (a)-(d), by combining with the empirical training error, all four generalization error bounds can be used to bound the empirical test error, but our bound is able to generate a much tighter upper bound. Such difference is mainly due to the fact that we replace the squared *gradient norm* in Li et al. (2020), the squared norm of

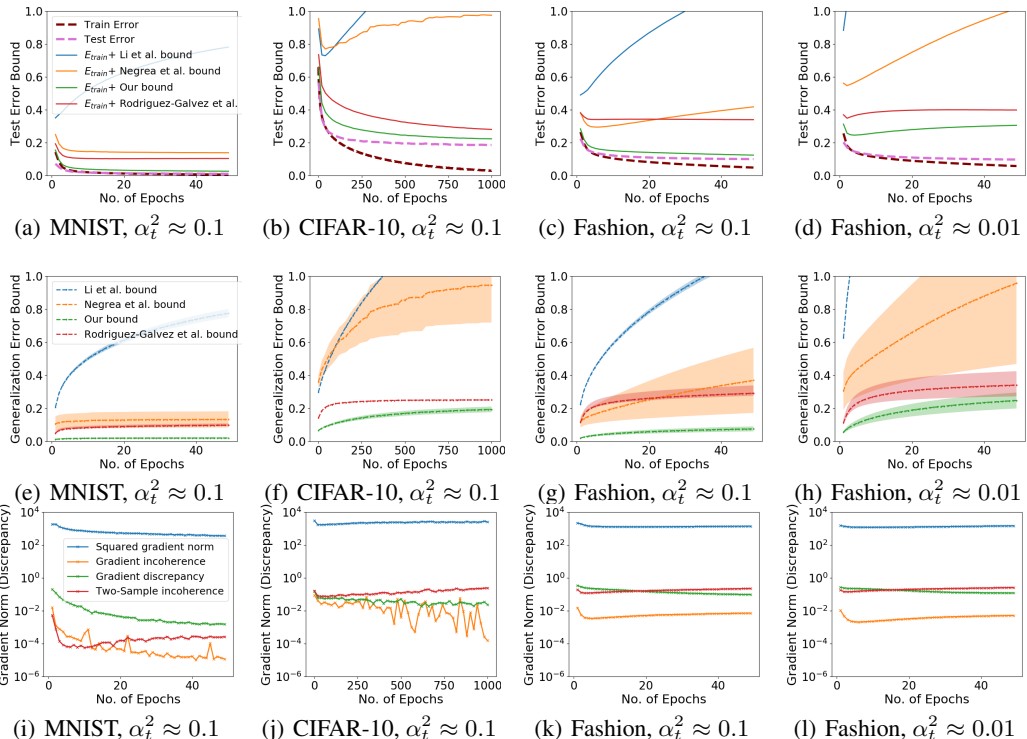

Figure 1: Numerical results for training CNN using SGLD ($\sigma_t = \sqrt{2\eta_t/\beta_t}$) on MNIST, Fashion-MNIST and CIFAR-10. X-axis shows the number of training epochs. (a)-(d) shows our bound is non-vacuous and can be used to bound the empirical test error. (e)-(h) compare our bound with the existing bounds and show the effect on $\alpha_t^2$. (i)-(l) show the key factors in each bound, i.e., the squared gradient norm in Li et al. (2020), the gradient incoherence in Negrea et al. (2019), the two-sample incoherence in Rodríguez-Gálvez et al. (2021), and the gradient discrepancy in our bound. Our bounds are numerically sharper than existing bounds, and larger $\alpha_t^2$ leads to tighter generalization bounds which is consistent with the theoretical analysis.

*gradient incoherence* in Negrea et al. (2019), and that of *two-sample incoherence* in Rodríguez-Gálvez et al. (2021) with the gradient discrepancy. Results in Figure 1 (e)-(h) show that our bounds are much sharper than those of Li et al. (2020) because our gradient discrepancy (Figure 1 (i)-(l)) is usually 2-4 order of magnitude smaller than the squared gradient norms appeared in Li et al. (2020). Our bounds are also sharper than those of Negrea et al. (2019) and Rodríguez-Gálvez et al. (2021) due to an improved dependence on $n$ from an order of $1/\sqrt{n}$ to $1/n$. Note that, even though the gradient incoherence in Negrea et al. (2019) is about 1 to 2 order of magnitude smaller than the gradient discrepancy for simple problems such as MNIST and Fashion-MNIST, the difference between the gradient incoherence and our gradient discrepancy reduces as the problem becomes harder (see results for CIFAR-10 in Figure 1(j)).

**Effect of Randomness.** Motivated by Zhang et al. (2017), we train CNN with SGLD on a smaller subset of MNIST dataset ($n = 10000$) with randomly corrupted labels. The corruption fraction varies from $0\%$ (without label corruption) to $60\%$. As shown in Figure 2 (d), for long enough training time, all experiments with different level of label randomness can achieve almost zero training error. However, the one with higher level of randomness has higher generalization/test error (Figure 2 (a) dashed lines). Our generalization bound also becomes larger as the randomness increases since the corresponding gradient discrepancy increases.

### 4.2 NOISY SIGN-SGD

**Optimization.** Figure 3 (a)-(d) show the training dynamics of Noisy Sign-SGD under various selections of $\alpha_t$. As $\alpha_t \to 0$, Noisy Sign-SGD matches both the optimization trajectory as well as the final test accuracy of the original Sign-SGD (Bernstein et al., 2018a). However, as $\alpha_t$ increases, the probability of getting 1 approaches 0.5, and $\xi_t$ approximates a uniform distribution. As a result, the corresponding Noisy Sign-SGD still converges, but the generalization performance is much worse.

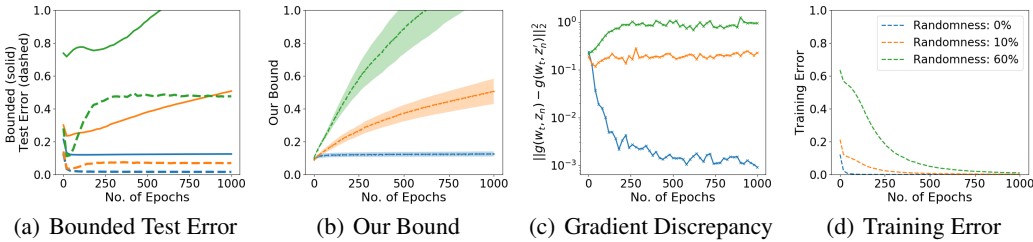

(a) Bounded Test Error     (b) Our Bound     (c) Gradient Discrepancy     (d) Training Error

Figure 2: Numerical results for training CNN using SGLD ($\sigma_t = 0.2\eta_t$) on a subset of MNIST ($n = 10000$) with different randomness on labels. (a) demonstrates that, as the randomness increases, the empirical test error (dashed lines) increases but still can be bounded by our generalization bound by combining the empirical training error (solid lines). (b) presents our bound in Theorem 2. (c) shows the gradient discrepancy $\|\nabla\ell(\mathbf{w}_t, z_n) - \nabla\ell(\mathbf{w}_t, z_n')\|_2^2$. (d) plots the training error. The gradient discrepancy increases as randomness increases, so does our generalization bound.

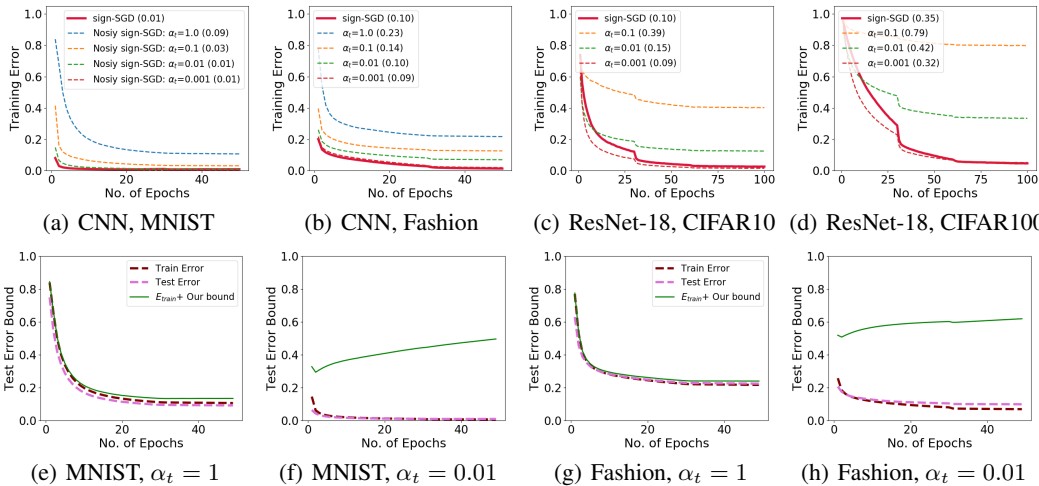

(a) CNN, MNIST    (b) CNN, Fashion    (c) ResNet-18, CIFAR10    (d) ResNet-18, CIFAR100

(e) MNIST, $\alpha_t = 1$    (f) MNIST, $\alpha_t = 0.01$    (g) Fashion, $\alpha_t = 1$    (h) Fashion, $\alpha_t = 0.01$

Figure 3: (a)-(d) show the training dynamics of CNN on MNIST and Fashion-MNIST, and ResNet-18 on CIFAR-10 and CIFAR-100 using noisy sign-SGD with different scaling $\alpha_t$. Legends indicate the choice of $\alpha_t$ and the numbers in brackets are test errors at convergence. As $\alpha_t \to 0$, Nosiy sign-SGD matches both the optimization trajectory as well as the final test accuracy of the original sign-SGD (Bernstein et al., 2018a). (e)-(f) show that empirical test error can be bounded by our bound and the corresponding training error. The larger $\alpha_t$ is the sharper our bound is.

**Generalization Bound.** Figure 3(e)-(f) show that our bound successfully bounds the empirical test error. The larger $\alpha_t$ is the sharper the upper bound is. However, larger $\alpha_t$ would slow down and adversely affect the optimization, e.g., Figure 3 (a)-(d) blue and orange lines. In practice, one needs to balance the optimization error and generalization by choosing a suitable scaling $\alpha_t$.

## 5 CONCLUSIONS

Inspired by recent advances in stability based and information theoretic approaches to generalization bounds (Mou et al., 2018; Pensia et al., 2018; Negrea et al., 2019; Li et al., 2020; Haghifam et al., 2020), we have presented a framework for developing such bounds based on expected stability for noisy stochastic iterative (NSI) learning algorithms. We have also introduced Exponential Family Langevin Dynamics (EFLD), a family of noisy gradient descent algorithms based on exponential family noise, including SGLD and Noisy Sign-SGD as two special cases. We have developed an expected stability based generalization bound applicable to any EFLD algorithm with a $O(1/n)$ sample dependence and a dependence on gradient incoherence, rather than gradient norms. Further, we have provided optimization guarantees for EFLD and establish such results for Noisy Sign-SGD and SGLD. Our experiments on various benchmarks illustrate that our bounds are non-vacuous and quantitatively much sharper than existing bounds (Li et al., 2020; Negrea et al., 2019).

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
