# OpenReview forum: "Stability based Generalization Bounds for Exponential Family Langevin Dynamics"
_ICLR.cc/2022/Conference — ICLR 2022 Submitted_

### Official Review · Reviewer_ipDu · 2021-10-27

**Correctness:** 4
**Technical Novelty And Significance:** 2
**Empirical Novelty And Significance:** 2
**Recommendation:** 5
**Confidence:** 2

**Main Review:**

This paper provides several technical improvements to a recent line of work that I am not familiar with. Thus I can hardly review that relevance  and novelty of these improvements (and I am thus putting a low confidence score). I am concerned by the fact that the article is hard to read in a self-contained way. The authors largely comment the articulation of their work with similar works, but almost never comment the relevance of their work for a wider community.

Another concern is that the article provides two improvements listed above, but I could not perceive why it was coherent to present them in a common article. This might add some confusion to the paper.

As a naive question, I would like to ask why expected stability bounds are preferred. It seems that they involve expectations that are hard to compute analytically, at best one can approximate these expectations by sampling. How should these bounds be used in the end? In practice, to choose stopping criteria for algorithms, for instance? Or in theory, to analyze the performance of some algorithms?

I found the generalization from Gaussian noise to exponential families novel and interesting.

In Section 2, I find difficult to determine what are your contributions and what was already known. For instance, Proposition 2 seems well-known right? Why not giving some references rather than proving it? Similarly, to what extent was Proposition 1 known?

Minor comments:
-  Section 2: "D be an unknown distribution OVER Z" (not "of")
- Why do you introduce the notation W_{0:(t-1)} = w_{0:(t-1)}? I am under the impression that this is twice the same object, but I might misunderstand.
- You sometimes use the notation KL(P,Q) and KL(P||Q).
- Conflict of notation between S_0 and S_n (for n=0!).



**Summary Of The Paper:**

This paper improves generalization bounds based on the notion of stability. The notion of stability measures the effect of resampling a data point to the output of the algorithm. The paper improves known analyses in two directions:
- It proposes to use the notion of expected stability rather than uniform stability, which should to be tighter.
- It generalizes the analysis to noises in an exponential family rather than Gaussian noise.


**Summary Of The Review:**

I am not confident with my review as this paper is quite technical and I am not familiar with this line of work. I would appreciate having more insights about the application of the new techniques and the relevance of exposing them together in this paper.

---

> ### Author Response · Authors · 2021-11-21
> **Response**
>
> 1. "I am concerned by the fact that the article is hard to read in a self-contained way. The authors largely comment the articulation of their work with similar works, but almost never comment the relevance of their work for a wider community."
>
> The reviewer raises a very good point. There has been fast-paced progress on the theme over the past few years, and our exposition in the paper was indeed 'from the trenches', with gory technical details, placing our work relative to recent advances, and illustrating the technical aspects we improve on. We have tried to summarize the related advances and our results at a somewhat high level in Section 1, and that is all we could do in a page-and-half.
>
> In recent years, SGLD and its variants are being widely used for training both discriminate and generative models. Many recent generative models, e.g., score-based methods [1], are based on sampling using Langevin dynamics. Variants of SGLD also get used in the context of differential privacy (DP), including DP SGD [2] and the exponential mechanism [3], where the mechanism requires analysts to sample from a pre-computed distribution which could be highly computational expansive.
>
> [1] Song, Yang, Jascha Sohl-Dickstein, Diederik P. Kingma, Abhishek Kumar, Stefano Ermon, and Ben Poole. "Score-based generative modeling through stochastic differential equations" ICLR, 2021.
>
> [2] Raef Bassily, Adam Smith, and Abhradeep Thakurta. Private empirical risk minimization: Efficient algorithms and tight error bounds. In 2014 IEEE 55th Annual Symposium on Foundations of
> Computer Science, pp. 464–473. IEEE, 2014.
>
> [3] McSherry, Frank, and Kunal Talwar. "Mechanism design via differential privacy." 48th Annual IEEE Symposium on Foundations of Computer Science (FOCS'07). IEEE, 2007.
>
> 2. "Another concern is that the article provides two improvements listed above, but I could not perceive why it was coherent to present them in a common article"
>
> In general, the goal here is to develop a tight generalization bound for a more general family of algorithms. For instance, our expected stability-based bound in Lemma 1 holds for a general family of noisy stochastic iterative algorithms and we apply it to get expected stability-based generalization bounds for EFLD in Theorem 2. We will add text at the beginning of Section 3 to make the connection between Section 2 and 3 more explicit and to address the concern.
>
> 3. "why expected stability bounds are preferred? How should these bounds be used in the end"
>
> The reviewer asks a very good question. At a high level, there is currently considerable focus on getting bounds that are both computable, possibly by Monte Carlo sampling (as remarked by the reviewer), and non-vacuous, i.e., less 1 for the probability of error in the context of classification. Note that the bound we present can be approximated by Monte Carlo sampling and our empirical results show that the bounds are non-vacuous. Further, note that all existing related work we cite have similar experiments.
>
> We share additional details regarding the bounds. Compare to the uniform stability approach which requires establishing a uniform bound over all samples, i.e., $\sup_{S_n, S_n^\prime} \sqrt{2 H^{2}\left(A(S_n), A\left(S_{n}^{\prime}\right)\right)}$,  expected stability only needs to bound the average, i.e., $E_{S \sim D^{n}} E_{z_{n}^{\prime} \sim D } \sqrt{2 H^{2}\left(A(S), A\left(S_{n}^{\prime}\right)\right)}$ which can be considerably sharper than the uniform bound, since we are replacing $\sup$ with $\mathbb{E}$. Besides, the uniform stability approach usually requires assuming bounded loss function and bounded gradient (Mou et al., 2018; Li et al., 2020), which leads to a bound depending on the Lipshitz constant.
>
> Note that test error can be bounded by the generalization error plus the training error and both of them are commutable along the training process, where usually training error decreases as we run more iterations and our generalization error bound increases mildly due to sum over small but non-negative terms per iteration. One can compute the sum of the training error and generalization bound to estimate an upper bound on the test error over iterations and pick the (optimal) number of iterations where the bound on the test error is minimized. Thus our bound can in fact serve as a stopping criterion.

---

> > ### Author Response · Authors · 2021-11-21
> > **Response Cont.**
> >
> > 4. " In Section 2, I find difficult to determine what are your contributions and what was already known"
> >
> > The propositions are essentially followed by definitions and reasonably simple analysis. Getting into the details, we did not quite find a direct reference to Proposition 2, yet it is a combination of existing pieces of a well-known result of Le Cam’s inequalities (Lemma 2.3 in [Tsybakov, 2008]), a result links the Hellinger distance to the Kullback divergence (Lemma 2.4 in [Tsybakov, 2008]) and Pinsker’s inequality (Lemma 2.5 in [Tsybakov, 2008]). The closest existing result to Proposition 1 is in Section 3.1 in [Mou et al. 2017], and we improve their result on uniform stability to expected stability. Another similar result is Theorem 7 in [Li et al, 2020], where they bound Bayes stability (not via Hellinger distance) by assuming the loss function is bounded, which is much more restrictive than our results.
> >
> > We have given complete proofs of all results we use, and details can be found in Appendix B. We will update Section 2 and add additional relevant references.

---

> > > ### Comment · Reviewer_ipDu · 2021-11-29
> > > **Thank you**
> > >
> > > I thank the authors for their answer. I am still unconfident with my review as this paper is quite technical and I am not familiar with this line of work. However, I would like to keep a grade marginally below the acceptance threshold as, from basic knowledge in theoretical machine learning and a reasonable effort, I was unable to perceive the stakes of this paper.  This opinion can easily be discarded if the other reviewers disagree.

---

### Official Review · Reviewer_j8ws · 2021-11-01

**Correctness:** 3
**Technical Novelty And Significance:** 3
**Empirical Novelty And Significance:** 3
**Recommendation:** 5
**Confidence:** 3

**Main Review:**

Strength.
1. The paper proves the generalization error of ''existing'' noisy stochastic iterative algorithms to have a $\mathcal O(1/n)$ sample dependence.
2. The exponential family noise is considered in the generalized SGD algorithms. The generalized error bound is based on expected stability, which seems to bring new ideas for general exponential family noise.  A thorough analysis of the error bound of exponential family noise SGD algorithms is provided.

Weakness.

1. The proposed algorithm does not show state-of-the-art performance.  The ultimate goal of the proposed algorithms is to achieve better performance on real data sets. Without strong support from the empirical study,  it is not convincible to believe why one bothers to study such more complicated cases.
2. Adding to the above point, the paper only reviews the SGD in the literature, there have been very recent works based on Langevin dynamics, e.g. replica-exchange Langevin dynamic SGD, second-order Hamiltonian Langevin dynamics. Those algorithms achieve state-of-the-art performance. It is questionable to ask if the newly proposed algorithms in this paper could provide better performance if the model architectures are improved.

**Summary Of The Paper:**

The paper under review considers the generalized SDLG algorithms with exponential family noise introduced to the mini-batch gradient descent. The analysis of the error bound is based on the control of the expected stability. Experiments are provided to support the proposed EFLD SGD algorithms.

**Summary Of The Review:**

The paper seems to provide a new family of SGD algorithms. The potential of the model seems to be broad. The theoretical analysis is recognized and deserves their own interest. However, the current state of the empirical study does not show promising performance.

---

> ### Author Response · Authors · 2021-11-21
> **Response**
>
> 1. 'it is not convincible to believe why one bothers to study such more complicated cases.'
>
> We would like to emphasize that the proposed EFLD is a framework for developing generalization bounds for noisy stochastic iterative (NSI) algorithms, and we consider an empirical comparison with noisy iterative gradient descent algorithms with generalization guarantees. Recent years have seen considerable progress in such algorithms with generalization guarantees, e.g., Li et al., Negrea et al., and we have empirically demonstrated that our proposed approach and analysis yields sharper bounds. Note that our guarantees hold on any datasets. SGLD (EFLD with Gaussian noise) shows the relevant sota performance (see Table 1) and is the only noisy iterative algorithm with generalization guarantees.
>
>
> Reviewer indirectly raises a broader point: why should one focus on establishing generalization bounds for (noisy iterative) algorithms when there are approaches with better performance on benchmark datasets? One key property of deep learning models is that they can fit any training data with (almost) 100\% accuracy on the training set ([Zhang et al. ICLR 2017]). However, such high accuracy (on the training set) does not imply anything for generalization performance. For example, one can take any of the existing benchmarks and make a fixed fraction of labels random. While the training set performance will still be good (Figure 2(d) of the main paper), the prediction performance will get progressively worse with increasing random labels (see dashed lines in Figure 2(a) of the main paper). Note that in real-world problems (not benchmarks), it is desirable to get a conservative but correct estimate of the generalization performance, rather than an optimistic and incorrect estimate
>
> [1] Zhang, Chiyuan, Samy Bengio, Moritz Hardt, Benjamin Recht, and Oriol Vinyals. "Understanding deep learning (still) requires rethinking generalization." ICLR 2017.
>
> 2. 'there have been very recent works based on Langevin dynamics, e.g. replica-exchange Langevin dynamic SGD... It is questionable to ask if the newly proposed algorithms in this paper could provide better performance if the model architectures are improved.'
>
> We would like to thank the reviewer for suggesting additional works (replica-exchange Langevin dynamic SGD, second-order Hamiltonian Langevin dynamics) based on Langevin dynamics.
> We would like to emphasize that this paper focuses on proposing a tighter generalization bound and extending Gaussian noise to general exponential family noise. However, the papers provided by the reviewer focus on the state-of-the-art performance regarding the optimization (performance on the training data), which is beyond the scope of this paper. We believe, most of the popular optimizers with proper hyper-parameter tuning can reach state-of-the-art performance. We have used the simplest optimizer SGD with momentum (M-SGD) to train a more complex model, i,e, ResNet-56 and it can beat SGHMC and reSGHMC as shown in the following table:
>
>
> Table 1: Test set performance comparison of the different optimization algorithms on CFIAR-10
>
> -------
>
> -- Model -- |  ---  M-SGD  ----    |   ---- SGLD ---- |        --- SGHMC* --|        ---  reSGHMC*  --|
>
> ResNet-20  |  $95.02\pm 0.23$| $93.55\pm 0.15$| $94.22 \pm 0.12$| $94.62\pm 0.18$ |
>
> ResNet-56 | $96.70\pm0.07$| $94.89\pm0.04$| $95.95\pm 0.10$| $96.12 \pm 0.06$ |
>
>
> Thus, we believe the reviewer's point of achieving state-of-the-art  **optimization** performance does not rule out the contribution of our paper, which is focusing on the **generalization performance**.
>
> Also, we would like to highlight that we have tried to replicate the results of replica-exchange Langevin dynamic SGD. We realized this algorithm has high computational complexity and cost. We believe comprising the computation efficiency in order to achieve optimizing performance is not desired in practice.

---

### Official Review · Reviewer_AX5S · 2021-11-04

**Correctness:** 4
**Technical Novelty And Significance:** 2
**Empirical Novelty And Significance:** 3
**Recommendation:** 5
**Confidence:** 3

**Main Review:**

There have been several works that generalize from gaussian settings to the more general exponential family, including the ones that are cited by the authors but also for example monotone retargetting and exponential family PCA. As such there are several tools for such an analysis that are pretty well known now. Other than that, the analysis follows closely with that of Li et al, with few minor but important differences.

I am not sure if the extension to exponential family is a major contribution. It allows for analysis of signSGD. But given several other works of similar nature, I am not sure if the analysis and the major contribution as claimed, including in the title of the paper justifies a publication.

It seems, however, that proposing gradient discrepancy itself is a useful contribution. Can the authors highlight the difference in the analysis from Li et al explicitly that led to this difference in the bounds ? Why is this sidelined as a side-effect? Does this follow from the new exponential family based analysis proposed here (it does not seem to be so) ? The experiments are also conducted based on this contribution. As such the main message/contribution of the paper and the empirical section seems disconnected. I am happy to be corrected.

**Summary Of The Paper:**

This paper provides generalization guarantees for exponential family sgld which is a more general setup than standard gaussian noise langevin dynamics as it allows for noise settings from any member of the exponential family. The authors point out that the setup they consider is a strict generalization of the standard langevin dynamics, and give another explicit example of sign sgd based generalization.

The authors use and show gradient discrepancy as opposed to gradient norms as used by Li et al, and gradient incoherence as used by Negrea et al. but the latter have 1/\sqrt{n}. Overall, from useful empirical observations it seems that the bounds proposed by the authors are tighter than these other bounds.

**Summary Of The Review:**

The main contribution of exponential family dynamics seems not very significant. Gradient discrepancy is sidelined as a minor contribution but is focussed on a lot more in the empirical section, which makes the paper seems disconnected.

---

> ### Author Response · Authors · 2021-11-21
> **Response**
>
> 1. "There have been several works that generalize from gaussian settings to the more general exponential family,... the analysis follows closely with that of Li et al, with few minor but important differences "
>
> We acknowledge the remark regarding the extensions from Gaussians to exponential families, including monotone retargeting and exponential family PCA. We are in fact intimately familiar with both of these results and can add references to the papers. Our current work, however, focuses on generalizing (stochastic gradient) Langevin dynamics which has literature spanning several decades. While existing work has focused on Gaussians (Weiner processes), we generalize the results to exponential families and establish stability-based generalization bounds for the entire family. Note that we compare our results to existing and recent related work and illustrate improvements both in theory and empirically. Further, even for the Gaussian case, our theoretical and empirical results are sharper than the existing results.
>
> 2. " I am not sure if the extension to the exponential family is a major contribution."
>
> We highlight a few specific novel and technical aspects of our work:
> - Existing work on Langevin dynamics, including the classical continuous cases and algorithms based on discrete updates, are based on Gaussians (Weiner process). Our paper is the first to generalize Langevin dynamics to general exponential families. The proposed exponential family Langevin dynamics (EFLD) generalizes the ideas in this context considerably, and the sharper analysis even improves the results for stochastic gradient Langevin dynamics (SGLD) both in theory and empirically.
>
> - Our proof is quite different from Li et al., 2020 which is based on the rotational invariance property of isotropic Gaussians. To be more specific, the core technical argument in [Li et al., 2020] for Gaussian Langevin dynamics, in particular, the proof of Lemma 21 is only applicable to Gaussians. The argument critically used rotational invariance of Gaussians, so that the problem can be reduced to 2 dimensions,  to avoid dimensional dependence.  In general, exponential families are not rotationally invariant and a similar analysis will end up picking up an undesirable dependence on the dimensionality. We build a new proof technique to bypass this issue. Our proof works for general exponential families, uses specific properties of exponential families, and does not pick up the dependence on the dimensionality. We give a brief overview of the technique in Section 3.3, and the detailed proof in Appendices C.2 and C.3. Our new proof technique not only applies to general exponential families, it in fact sharpens the result for the Gaussian case (SGLD) with dependence on gradient discrepancy rather than gradient norm and a $O(1/n)$ sample dependence.
>
> - In practice, one can only implement finite precision gradient descent. One will not be able to implement continuous Gaussian noise with infinite precision in practice. Exponential family Langevin dynamics allows us to work with such finite precision versions and provide guarantees.

---

> ### Author Response · Authors · 2021-11-21
> **Response Cont.**
>
> 3. "Can the authors highlight the difference in the analysis from Li et al explicitly that led to this difference in the bounds"
>
> We are happy to discuss why the gradient discrepancy shows up and the difference with the analysis in [Li et al.]. We did not mean to sideline this aspect of our result -- in fact, the dependence on gradient discrepancy is one of the reasons our bound is (empirically) sharper, e.g., see Figure 1(i), (j), (k), (l). We also note that [Negrea et al., 2019] has a related term called gradient incoherence, but they get a $O(1/\sqrt{n})$ sample dependence.
>
> [Li et al., 2020] approach the analysis from a PAC-Bayes perspective and define a prior involving a "zero data point" (discussion after Lemma 10 in [Li et al.,]: ``Define $P=E_{\bar{S} \sim D^{n-1}}[Q_{(\bar{S},\mathbf{0}})]$, where $0$  is the zero data point (i.e., $F(w,\mathbf{0}) = 0$ for all $w$. The way the "zero data point" is defined, the gradient $\nabla_{w} F(w,\mathbf{0}) = \mathbf{0}$. The second term in the KL-divergence in the analysis of [Li et al.] has this "zero data point", which has zero gradient. As a result, they end up getting a term only depending on the norm of the gradient of the data points.
> Note that, the "zero data point" needs $F(w,\mathbf{0}) = 0$ for all $w$, where $F(w,z)$ is the surrogate loss function, $z$ is an instance, e.g., $z_i = (x_i,y_i)$. If $F$ is the log-loss/cross-entropy loss for a classification problem, it is unclear how one would get such a "zero data point". This aspect of their analysis seems to stem from the PAC-Bayes perspective, in particular the need to construct a prior.
>
> In contrast, our analysis is based on expected stability, where the second term in the Hellinger divergence and subsequently the KL-divergence includes a different data point $z'$, not the "zero data point." Our proof technique (Appendix C.2) starts with a first-order Taylor expansion, which introduces the natural parameter difference which is equivalent to the gradient discrepancy, and the analysis handles all terms suitably so that the final bound is in terms of the gradient discrepancy term. As we show empirically in Figure 1(i), (j), (k), (l), the norm of the gradient discrepancy is 1-2 orders of magnitude smaller than the norm of the gradient.
>
> In Remark 8 of [Li et al., 2020], a comparison is made with the concept of uniform stability. They correctly point out that uniform stability is distribution independent whereas their treatment based on PAC-Bayes, rather than Bayes stability, is distribution dependent. Our work focuses on expected stability, rather than uniform stability, which makes our bound distribution dependent and addresses the concern in Remark 8 of [Li et al., 2020].

---

### Official Review · Reviewer_4ShW · 2021-11-04

**Correctness:** 4
**Technical Novelty And Significance:** 3
**Empirical Novelty And Significance:** 3
**Recommendation:** 8
**Confidence:** 4

**Main Review:**

- Using the generalization bound in Theorem: In proposition 1, the authors provide a generalization bound based on the distributional stability in terms of Hellinger distance. However, in most of the results in the paper they mainly use the upper bound based on KL divergence (Prop. 2). Is that correct?

- The dependence of the learning rate on the optimization trajectory: In your bound alpha_t depends on the dataset and trajectory. It basically depends on some sort of Lipschitz constant of the loss function. For instance in SGLD, alpha_t depends on the learning rate and inverse temperatures. Therefore, I think it can limit us to use very small learning rates. Also, how did you make sure that this constraint is satisfied in your numerical studies?

- The comparison with Rodríguez-Gálvez et al.: Your bound is very similar to the generalization bound in Haghifam et al'2020. As you mentioned the bound in Haghifam et al'2020 is stated for LD algorithm. Recently, the bound in Haghifam et al'2020 is extended to SGLD in [1]. The bound in Rodríguez-Gálvez et al has worse dependence on n (1/sqrt(n)), however, they have a decaying factor in their bound to compensate using of the chain rule. It would be nice to numerically compare your bound with the bound in [1].

- Applying your framework for analyzing SGD:  What are the roadblocks for applying your bound to analyzing SGD?

- Surrogate loss: I think you should distinguish the loss function that we use to measure the generalization error with the loss function that is used for training. You should use different notations.


[1] "On random subset generalization error bounds and the stochastic gradient langevin dynamics algorithm", Borja Rodríguez-Gálvez, Germán Bassi, Ragnar Thobaben, Mikael Skoglund


**Summary Of The Paper:**

This paper studies the generalization of iterative noisy learning algorithms. First of all, the authors provide a new generalization bound based on the Hellinger distance. This bound is basically based on the Hellinger distance between the output of the algorithm when one of the points in the training set is replaced with a fresh sample. They they use this bound to provide a generalization error guarantee for a broad class of noisy iterative algorithm which is defined as W_t = W_{t-1} - eta_t, where eta_t conditioned on the past iterated follows an exponential family distribution which for instance include SGLD. For this broad class, they provide a unified generalization bound that includes SGLD and Noisy Sign-SGD. Finally, the authors provide numerical study of their bound for several benchmarks and show that their bound achieves better estimate of generalization error.

**Summary Of The Review:**

Very interesting and solid contribution!

---

> ### Author Response · Authors · 2021-11-21
> **Response**
>
> 1. 'in most of the results in the paper they mainly use the upper bound based on KL divergence (Prop. 2). Is that correct?'
>
> That is correct. The generalization error can be bounded by the Hellinger divergence as proposed in proposition 1. The Hellinger divergence can be bounded by the KL divergence (Proposition 2). So, in this paper, we focus on bounding the KL divergence. Two addition remarks: first, note that Hellinger divergence does not require absolute continuity, whereas KL divergence does, so in the future, a more flexible and possibly sharper analysis may be possible by directly bounding the Hellinger divergence; and second, Hellinger divergence can be bounded by the TV (total variation) divergence (well known, also in proof of Proposition 2), which can provide alternative ways of proving bounds.
>
> 2. 'The dependence of the learning rate on the optimization trajectory'
>
> The bound on $\alpha_t$ is a data-dependent and trajectory dependent measure of max gradient discrepancy. Note that $\alpha_t$ which can be computed during the training process, and it gives much more relaxed upper bound on the step size compared to those in the related work (Mou et al., 2018; Li et al., 2020, Hardt et al. 2016). For SGLD, Mou et al., 2018; Li et al., 2020 require step size to be bounded by $\frac{\sigma_{t}}{L}$, where $L$ is the global Lipschitz constant.  Lemma 1 in our work shows that the step sizes need to be bounded by $\frac{\sigma_{t}}{\Delta_{t}(\bar{S})}$, which is considerably more relaxed since $\Delta_{t}(\bar{S})$ is much smaller than the global Lipschitz constant $L$, which is a uniform bound over the whole parameter space.  Also, Hardt et al. (2016) require $1/t$ step sizes, which is quite small.
> Further, one would expect $\Delta_{t}(\bar{S})$ to decrease as training proceeds since the gradients shrink as the loss function is minimized. Thus, the constraint on step size does not require the step sizes to be as small as $\sigma_t/L$ or $1/t$.
>
> We mention that $\alpha_t$ is a data-dependent quantity that can be computed in the experiments.  We numerically verified that the learning rates used in our experiments satisfy the conditions in the theory.
>
> 3. 'The comparison with Rodríguez-Gálvez et al.'
>
> We have updated Figure 1 in our paper to include additional results comparing our bound with the bound provided by Rodríguez-Gálvez et al. In our experiments, we consider $\phi=\frac{1}{2}(1+\text{erf}(x))$ used in Haghifam et al. to compute the estimate $\pi_{J,t}$ appeared in equation (8) (Proposition 6 in Rodríguez-Gálvez et al.).
> Overall, the new CMI bound in Rodríguez-Gálvez et al., is tighter than the bound proposed by Negrea et al., and Li et al., because of the additional decaying factor, and such observation is consistent with the observation made by Haghifam et al. However, our bound is still the tightest among all due to the dependence on $\frac{1}{n}$ rather than $ \frac{1}{\sqrt{n}}$.
>
> 4. 'What are the roadblocks for applying your bound to analyzing SGD?'
>
> The current analysis bounds the KL divergence between two distributions over the parameters, which can be analyzed in terms of Le Cam Style Divergence (LSD). It is not straightforward to extend this type of analysis to SGD since the current does need absolute continuity of the distributions. Besides, Hardt et al. (2016) have established uniform stability bounds for SGD for various types of problems and showed that without extra noise, the uniform stability bound of SGD can be worse than SGD with additive noise, although these are all upper bounds. If we have to speculate, characterizing the SGD noise (due to random min-bathes) may help build a similar analysis based on the local properties of the training trajectory.
>
> 5. 'Surrogate loss'
>
> Thanks for making this point. Our analysis is for the surrogate loss which is used for the optimization and which serves as an upper bound to the true risk. Bounding the expected surrogate loss implies a bound on the true risk. We will add a remark in the paper to clarify this aspect. All our experiment uses cross-entropy loss (surrogate loss) for both training and evaluating generalization error.

---

> > ### Comment · Reviewer_4ShW · 2021-11-27
> > **Thanks**
> >
> > I would like to thanks the authors for their response. The updated revision includes my main questions and concerns.

---

### Author Response · Authors · 2021-11-21
**General Response to all Reviewers**

We thank the reviewers for their thoughtful comments! We are delighted to hear that reviewers find the problem we tackle to be interesting (Reviewer 4ShW) and our approach of generalization from Gaussian noise to exponential families novel (Reviewer ipDu, Reviewer j8ws, Reviewer 4ShW), the expected stability and gradient discrepancy to be useful (Reviewer AX5S). We address the most important/common questions here and answer specific questions in our individual responses to each Reviewer.

**Novelty, significance, and contributions**
1. Existing work on Langevin dynamics, including the classical continuous cases and algorithms based on discrete updates, are based on Gaussians (Weiner process). Our paper is the first to generalize Langevin dynamics to general exponential families.

2. Our core technical proof is quite different from Li et al., 2020, which is based on the rotational invariance property of isotropic Gaussians. Using the rotation invariance property, Li et al., 2020 represented $p$-dimensional Gaussians in 2-dimensions. Without this important 2-d representation step, their analysis would have picked up an undesirable dependence on the dimensionality. The rotational invariance property does not hold for general exponential family distributions, so the Li et al., 2020 proof technique does not work in the general case. Our technical proof (Appendix C.2) keeps the analysis and integral in the original $p$-dimensions and has four steps: a first-order Taylor, removing distributional dependence of the intermediate parameter obtained from Taylor, bounding the density ratio from the Le-Cam style divergence, and finally bounding the $p$-dimensional integral in terms of the gradient discrepancies while avoiding the dimensional dependence. Throughout the analysis, we only use the definitions and properties of exponential family distributions and convex functions.

3. Our bound is sharper than exiting bounds in the following aspects. First, we replace norms on gradients, e.g., in Mou et al., 2018, Li et al., 2020, with gradient discrepancy which are quantitatively much smaller as we show in Figure 1(a) and (e). Second, our bound has a $O(1/n)$ sample dependence instead of $O(1/\sqrt{n})$ (Negrea et al. 2019;  Haghifam et al., 2020; Rodriguez-Galvez et al. 2021). Third,  our bound relaxes the effective upper bound on step size compared to existing bounds  (Mou et al., 2018;  Li et al., 2020) from $\frac{\sigma_t}{L}$ to $ \frac{\sigma_{t}}{\Delta_{t}(\bar{S})}$, where $L$ is the global Lipschitz constant and $\Delta_t(\bar{S})$ is local data-dependent measure of gradient discrepancy. Thus, our bound in Theorem 3 inherits the strengths of the existing bounds with dependence on norms of gradient discrepancy, no dependence on L, $O(1/n)$ sample dependence, and flexible data-dependent choices for $\eta_t$.

4. In practice, one can only implement finite precision versions of gradient descent (GD), which have been shown to be robust to even low precision versions, including the extreme case of sign SGD. In particular, one is not able to implement continuous Gaussians with infinite precision in practice. Exponential family Langevin dynamics allows us to work with noise with finite-precision representations and still do a rigorous analysis. Note that some related work has been started in the context of differential privacy [1].

[1] C. Canone, G. Kamath, and T. Steinke, The Discrete Gaussian for Differential Privacy, NeurIPS, 2020.

**We have made the following updates in the paper:**

1. As suggested by Reviewer 4ShW, we have updated Figure 1 in our main paper to include additional results comparing our bound with the bound provided by Rodríguez-Gálvez et al. The experimental results show our bound is still the tightest among all the compared methods.

2. To address Reviewer 4ShW 's concern about the condition on $\alpha_t$, we have added  Remark 3.3 following our main theorem (Theorem 2). The condition on $\alpha_t$ is much more benign compared to related work and it doesn’t require a very small learning rate.

---

### Decision · Program_Chairs · 2022-01-20

**Decision:**

Reject

**Comment:**

This paper focuses on generalization bounds for exponential family Langevin dynamics, which extends related recent work for stochastic iterative algorithms such as SGLD in several ways. They derive expected stability bounds for a more general class of noisy stochastic iterative algorithms, leading to an exponential family variation of Langevin dynamics and a noisy version of the sign-SGD algorithm. The contributions are technical and quite positively received by one reviewer, while the others were not convinced to change their opinions during the author response as there were concerns on the limitation of the theoretical contributions and the extent to which these contributions have implications on achieving state of the art performance. While I find it valid that the scope of the paper focuses on generalization bounds and provide improvements over the existing literature, rather than on empirical benchmarks or on optimization-related aspects, the overall borderline impression of the reviewers on the whole suggests that a refined version of the paper that further clarifies the contributions and makes clear its impacts as well as limitations may make for a stronger and more impactful paper.